# The Cause of Hereditary Hearing Loss in *GJB2* Heterozygotes—A Comprehensive Study of the *GJB2*/DFNB1 Region

**DOI:** 10.3390/genes12050684

**Published:** 2021-05-01

**Authors:** Dana Safka Brozkova, Anna Uhrova Meszarosova, Petra Lassuthova, Lukáš Varga, David Staněk, Silvia Borecká, Jana Laštůvková, Vlasta Čejnová, Dagmar Rašková, Filip Lhota, Daniela Gašperíková, Pavel Seeman

**Affiliations:** 1Neurogenetic laboratory, Department of Paediatric Neurology, 2nd Faculty of Medicine, Charles University and University Hospital Motol, 15006 Prague, Czech Republic; anna.meszarosova@lfmotol.cuni.cz (A.U.M.); petra.lassuthova@fnmotol.cz (P.L.); david.stanek@lfmotol.cuni.cz (D.S.); pavel.seeman@lfmotol.cuni.cz (P.S.); 2Department of Otorhinolaryngology–Head and Neck Surgery, Faculty of Medicine and University Hospital, Comenius University, 85107 Bratislava, Slovakia; varga.lukas@gmail.com; 3Diabgene Laboratory, Institute of Experimental Endocrinology, Biomedical Research Center, Slovak Academy of Sciences, 84505 Bratislava, Slovakia; silvia.borecka@savba.sk (S.B.); daniela.gasperikova@savba.sk (D.G.); 4Department of Medical Genetics, Masaryk Hospital in Usti nad Labem, Regional Health Corporation, 40011 Ústí nad Labem, Czech Republic; jana.lastuvkova@kzcr.eu (J.L.); vlasta.cejnova@kzcr.eu (V.Č.); 5Centre for Medical Genetics and Reproductive Medicine GENNET, 17000 Prague, Czech Republic; dagmar.raskova@gennet.cz (D.R.); filip.lhota@gennet.cz (F.L.)

**Keywords:** hearing loss, *GJB2* monoallelic variant, DFNB1 region, next generation sequencing

## Abstract

Hearing loss is a genetically heterogeneous sensory defect, and the frequent causes are biallelic pathogenic variants in the *GJB2* gene. However, patients carrying only one heterozygous pathogenic (monoallelic) *GJB2* variant represent a long-lasting diagnostic problem. Interestingly, previous results showed that individuals with a heterozygous pathogenic *GJB2* variant are two times more prevalent among those with hearing loss compared to normal-hearing individuals. This excess among patients led us to hypothesize that there could be another pathogenic variant in the *GJB2* region/DFNB1 locus. A hitherto undiscovered variant could, in part, explain the cause of hearing loss in patients and would mean reclassifying them as patients with *GJB2* biallelic pathogenic variants. In order to detect an unknown causal variant, we examined 28 patients using NGS with probes that continuously cover the 0.4 Mb in the DFNB1 region. An additional 49 patients were examined by WES to uncover only carriers. We did not reveal a second pathogenic variant in the DFNB1 region. However, in 19% of the WES-examined patients, the cause of hearing loss was found to be in genes other than the *GJB2*. We present evidence to show that a substantial number of patients are carriers of the *GJB2* pathogenic variant, albeit only by chance.

## 1. Introduction

In neonates, hearing loss is the second most common sensory deficit after color blindness, with an incidence of 1:500–1000 [1]. The causes of hearing loss are very heterogeneous, but if we exclude non-hereditary causes, such as infections and premature births, the remaining genetic causes account for almost 80% of hearing loss in patients. Even genetic causes are very diverse; hearing loss is caused by pathogenic variants in more than 110 genes, but the most frequent type of inheritance is the autosomal recessive hearing loss (referred to as the nonsyndromic, autosomal-recessive deafness—DFNB) [2]. In addition, almost 40% of Czech patients with autosomal recessive hearing loss have pathogenic biallelic variants in the *GJB2* gene (OMIM 121011), causing the DFNB1 type of deafness [3]. However, a small number of DFNB1 patients only have one heterozygous pathogenic (monoallelic) *GJB2* variant. Additionally, in 2014, Chan et al. published a meta-analysis of 48 case-control studies [4] and confirmed that the frequency of heterozygotes for a truncating *GJB2* variant (monoallelic) among patients with hearing loss is twice as high when compared to the normal-hearing population.

The most common cause of hearing loss detected worldwide is the c.35delG pathogenic variant. This variant has frequently been identified in European, North African, Middle Eastern, Asian, and American populations [5]. In the Czech Republic, among patients with hearing loss, the c.35delG variant accounts for 82.8% of all pathogenic *GJB2* variants (3). Furthermore, in our previous study, we showed that the frequency of c.35delG heterozygotes in Czech patients with hearing loss is 4%, and the figure is twice as high when compared to a frequency of 1.8% in normal-hearing persons with the monoallelic (heterozygous) c.35delG variant, as presented in the public database GnomAD for the non-Finnish European (NFE) population [6,7].

Therefore, we hypothesized that some of the patients may be explained by a combination of known monoallelic and some novel pathogenic *GJB2* variant, presumably a small copy number variant (CNV), to compensate for the frequency difference. A similar idea was suggested by del Castillo et al., who discovered a large deletion outside the *GJB2* region. The deletion, del(GJB6-D13S1830), is frequent among patients in Spain, France, the United Kingdom, Israel, and Brazil, and could explain the hearing loss in a large number of patients with a heterozygous pathogenic *GJB2* variant [8]. In addition, an infrequent large deletion, del(GJB6-D13S1854), was recently reported [9]. Nevertheless, both of the large deletions are very rare among Czech patients [10].

Seven deletions in the DFNB1 (13q11-q12) region have been reported worldwide. In the region outside of the *GJB2* gene, an intersection of five of these deletions displays a common interval; this is where the cis regulatory element is probably located, which is important for the regulation of *GJB2* gene expression (Figure 1) [9,11,12,13,14]. The presence of the cis regulatory element has been suggested by studies showing a reduction of *GJB2* expression. However, the deletion does not affect the gene per se [15,16,17,18]. Therefore, we focused on a continuous analysis of the DFNB1 (13q11-q12) region inclusive of the *GJB2* gene and the common interval (Figure 1). For the analysis, and in order to cover CNVs in a resolution range below the discrimination of chromosomal microarray technology, we used the next-generation sequencing (NGS) technology.

Our study comprises two parts: The first part examined the second expected pathogenic variant in the DFNB1 (13q11-q12) region, and for that we initially examined 28 patients by a continuous NGS of this region, which covered 0.4 Mb. The comprehensive NGS data analysis included the CNV analysis that used different software to detect the possible deletion as well as the presence of small-scale variants. In the second part, we supposed that a number of patients are only carriers of one heterozygous pathogenic *GJB2* variant by chance, and that the cause of their hearing loss is on a gene other than the *GJB2.* Therefore, we performed whole exome sequencing (WES) of 49 patients, 17 of whom we examined in both parts of the project.

## 2. Materials and Methods

The patients selected for this study came from two laboratories dedicated to the genetic testing of hearing loss: the University Hospital Motol in the Czech Republic and the Biomedical Research Center in Slovakia. Patients in whom the non-genetic causes of hearing loss were excluded (severe prematurity, perinatal risks, very low birth weight, meningitis, etc.) were referred by geneticists in order to be investigated for a genetic cause of hearing loss. All patients were unrelated, and the majority of them presented with early/prelingual hearing loss and fulfilled the criteria for *GJB2* testing (bilateral, nonsyndromic, prelingual or early hearing loss, and sensorineural). Patients’ parents had normal hearing, and an autosomal recessive mode of inheritance was considered the most probable cause. The *GJB2* gene (NM_004004.5 coding exon 2 and non-coding exon 1) was examined before the study, and patients with only one heterozygous (monoallelic) pathogenic variant were selected. Prior to the study, the large deletions, del(GJB6D13S1854) and del(GJB6D13S1830), were excluded with PCR or MLPA. The ethics committees of the University Hospital Motol, Prague, Czech Republic and the University Hospital, Bratislava, Slovakia approved the study.

For the examination of the DFNB1 (13q11-q12) region, a custom-design SureSelect Target Enrichment kit (Agilent technologies, Santa Clara, CA, USA) was used to continuously cover this region (hg19 reference genome–chr13: 20 697 953-21 105 890). We examined the genomic DNA of 31 Czech patients from which 28 were heterozygotes for pathogenic *GJB2* variant and three, that were homozygotes for the c.35delG (*GJB2*, NM_004004.5) variant. These homozygotes, that carry the biallelic pathogenic *GJB2* variant, were used as “negative” controls since it was expected that they would not carry the unknown mutation.

For the haplotype analysis, 29 SNPs from the NGS data of the DFNB1 (13q11-q12) region were selected for the assembly of the haplotype for the *GJB2* region (chr13: 20 698 553–20 871 892). The WES of 49 patients (29 Czech and 20 Slovak patients) was examined with a Human All Exon V6 kit (Agilent Technologies, Santa Clara, CA, USA). Seventeen of the 49 patients were analyzed previously for the DFNB1 (13q11-q12) region.

Both libraries were sequenced on Hi-Seq platforms (Illumina, Inc., San Diego, CA, USA). The small sequence changes (single-nucleotide variants and indels) in the NGS data were analyzed with the SureCall (Agilent technologies, Santa Clara, CA, USA) and Genome Analysis Toolkit (GATK) software. Only variants with a frequency of <1.0% in public databases (GnomAD, ExAC) were evaluated. The pathogenicity of the variants was evaluated according to the following criteria: ACMG criteria modified for genetic hearing loss [19], previously published pathogenic variants in HGMD professional, the values of prediction programs (SIFT, Mutation Taster, PolyPhen2), and conservation scores.

The larger sequence variants (CNVs) in the NGS data were analyzed differently for the DFNB1 region, and the WES was based on our own software testing for suitability to each platform. The copy number variant (CNV) analysis of the DFNB1 region was performed with NextGene (Softgenetics, State College, PA, USA) and Meta-SV analyses. The Meta-SV analysis tool [20] was used to detect and aggregate CNV in multiple tools. The tools used during the Meta-SV analysis were the following: BreakDancer [21], CNVnator [22], BreakSeq2 [23], Pindel [24], Manta [25], Lumpy [26], WHAM [27], and CNVkit [28]. Only CNV variants that were detected in heterozygous patients and not present in the homozygous controls were evaluated.

The copy number variant (CNV) analysis of the WES was performed with GATK Best Practices for Germline Copy Number Variation (available at: https://github.com/broadinstitute/gatk/tree/master/scripts/cnv_wdl/germline, accessed on 9 September 2020) in two steps. In the first step, a reference model was prepared from 40 healthy control WES samples (BAMs). In the next step, the reference model was used to perform CNV calling in a cohort of patients (BAMs).

The detected variants were verified using Sanger sequencing that included the small deletions (less than 130 bp). The primers used for the verification of the deletion were placed at a sufficient distance (around 200 bp) from the deletions’ start/end in order to prevent possible allele drop out when the primer, or its part, was placed in the deleted region. Where possible, a segregation analysis of the detected variants was conducted within the family. A deletion larger than 3000 bp was verified by quantitative–comparative fluorescent PCR (QCF PCR) and used to compare a fragment from the deleted region with a reference fragment from a non-deleted region on another chromosome.

As population-relevant controls, 3278 gamete donors from the Centre for Medical Genetics and the Reproductive Medicine Gennet were analyzed for the carrier frequency of the c.35delG variant in the *GJB2*. The data were obtained from the results of a routine pre-conception genetic screening for autosomal recessive disorders. Written informed consent for the use of personal particulars as well as genetic information for research purposes was obtained from the individuals.

## 3. Results

Using two different approaches of NGS, we examined the DFNB1 (13q11-q12) region in 28 patients and the WES in 49 patients—17 of whom were examined by both approaches.

### 3.1. Continuous Examination of the DFNB1 (13q11-q12)

The first part of the project included 28 patients with different heterozygous (monoallelic) pathogenic *GJB2* variants, with a predominance of the c.35delG variant (Figure 2). The analysis of the DFNB1 continuous region showed six possible copy number variants (deletions) detected by two different types of software—NextGene and Meta-analyses (Figure 2). Each software type predicted different deletions without any overlap. The NG1 deletion (chr13:20 826 120–20 829 603), the only one detected by NextGene, was confirmed in three patients in the heterozygous state with QCF PCR. The longer CNV deletion (gssvL29055), which overlaps with NG1, is present in the Database of Genomic Variants (DGV, accessed 02/2021) with a reported frequency of 9.07%; it was also detected in 439 out of 4842 individuals [29]. Furthermore, an NG1 deletion was examined and confirmed in one patient from the WES in the heterozygous state; one patient was a homozygote. Additionally, we tested the NG1 in 76 healthy individuals and confirmed one homozygote and six heterozygotes.

Another five deletions were detected by Meta-analyses. The deletions DS1–DS4 were detected in three patients, and the deletion DS5 in four patients. The length of the deletions (15–130 base pairs) was suitable for verification by Sanger sequencing; however, none of these five deletions (DS1–DS5) were confirmed by Sanger sequencing.

Furthermore, in an NGS data analysis of single nucleotide variants (SNVs), we only detected the c.-22-6 *GJB2* variant in patient 218517, which was also later confirmed by WES; however, no other pathogenic variant was found. We focused on the *GJB2* variants listed in the HGMD professional (HGMD, accessed 02/2021) with a special interest in the regulatory and splicing regions, yet no disease-causing variant was detected in our samples.

### 3.2. Haplotype Analysis of the DFNB1 (13q11-q12)

A haplotype analysis of the 28 patients and the three controls did not reveal any shared core haplotype among the patients (Appendix A). Therefore, it was not possible to identify a group of patients in whom a common pathogenic unknown variant was present.

Only the common general haplotype among the controls and patients was detected (chr13 g:20 761 888-20 800 039).

### 3.3. WES

In the second part of the project, the WES analysis of 49 patients (29 Czech and 20 Slovak patients), with different heterozygous (monoallelic) pathogenic GJB2 variants, detected possible pathogenic variants in 11 patients (22%). The specific variants and genes are summarized in Appendix A. We detected variants in different genes, but intronic variants were surprisingly detected in the *GJB2* gene of two patients, which offered a possible explanation for the patients’ hearing loss; these variants are in the UTR5´region of *GJB2*—the c.-22-2 and c.-22-6. The hearing loss in four other patients was caused by pathogenic biallelic variants in the well-known hearing loss-causing genes: *STRC* (OMIM 606440); *MYO15A* (OMIM 602666); *SLC26A4* (OMIM 605646); and *LOXHD1* (OMIM 613072). Additional variants were detected in five rarely diagnosed genes. In the *GPSM2* gene (OMIM 609245, NM_001321039.1), the homozygous c.858_859delinsGT, p.(Tyr286*) variant was detected in a patient with profound hearing loss. The patient also had hydrocephalus and a dilated ventricular system, although these were not initially connected to hearing loss. In the *MPZL2* gene (OMIM604873, NM_005797.3), the homozygous variant c.72del, p.(Ile24Metfs*22) was detected in one patient. In the *POU3F4* gene (OMIM300039, NM_000307.4), one male patient with profound hearing loss and normal-hearing parents was found to be hemizygote for the c.607_610del, p.(Gln203Glufs*37) variant. In the *SLITRK6* gene (OMIM609681, NM_032229.2), the variant c.1240C>T, p.(Gln414*) in the homozygous state was detected in a patient who suffered from hearing loss, was myopic from two and a half years of age, and needs to wear glasses. In several patients, different variants of uncertain significance (VUS) were detected (Appendix A). From the VUS-classified variants, the variants that stand out are in the *KARS1* gene (OMIM601421, NM_001130089.1). The variants c.871T>G, p.(Phe291Val) and c.881T>C, p.(Ile294Thr), which are both classified as VUS, were detected in a patient with autosomal recessive hearing loss; both are in the trans position and are predicted to affect the splicing. Even without the two patients found to have the second pathogenic variant in the non-coding part of the *GJB2*, the diagnostic yield of the WES in patients is high. In other words, a hearing loss of 19% in monoallelic patients (9/47) was due to variants in genes other than the *GJB2* (Figure 2), while these *GJB2* heterozygotes are carriers only by chance. For the WES examination, patients with different pathogenic heterozygous *GJB2* variants were selected, although the most frequent variant, c.35delG, was present in the majority (61%, 30/49) of the examined patients (Figure 2). Moreover, the diagnostic yield of the WES among c.35delG heterozygotes was almost the same at 20% (6/30). From among the 17 patients examined in both parts of the project (DFNB1 and WES), the cause of hearing loss in 4 of them was found by WES.

## 4. Discussion

The problems and limitations of diagnosing hereditary hearing loss have been known for some time due to the fact that there is a higher prevalence of individuals with a heterozygous pathogenic *GJB2* variant among patients with hearing loss than there is among the normal-hearing population [4]. In our study, we aimed to detect the cause of hearing loss in patients with only one pathogenic *GJB2* variant. In a comprehensive study of the DFNB1 (13q11-q12) region among Czech patients with hearing loss, we did not find the pathogenic variant CNV, nor the SNV in the DFNB1 locus. However, WES revealed the cause of hearing loss in 22% of patients.

The DFNB1 analysis, performed with different software types, revealed six CNV deletions, but we were only able to verify the NG1 deletion with routine genetic methods. The overlapping of the NG1 deletion with the frequent deletion (gssvL29055) suggested convincingly that this variant, which does not interfere with the common interval, is not pathogenic. For the analysis of the DFNB1 region, we chose NGS technology because the resolution for detecting CNV is below the range of chromosomal microarray technology. Despite the fact that NGS is widely used in genetic diagnostics, the CNV analysis from the NGS data is often limited, while chromosomal microarray technology and multiplex ligation-dependent probe amplification (MLPA) are the golden standards for CNV detection [30]. To reduce these limitations, we combined several software types to investigate the region fully. Our results showed that there is no common deletion or pathogenic variant in the DFNB1 region in Czech patients. Therefore, we inferred that there was no variant that would presumably, in combination with the already known *GJB2* pathogenic variant, explain the cause of hearing loss in a substantial number of patients with only one heterozygous pathogenic *GJB2* variant.

In the group of 49 patients investigated by WES, the diagnostic yield was 22%, which is higher than the results of our previous study of patients without pathogenic *GJB2* variants and tested by NGS. Last year, we published a study where 21% of 197 Czech patients with hearing loss, and where *GJB2* pathogenic variants had already been excluded, were clarified by a NGS gene panel. [31]. The diagnostic yield of the WES in patients with hereditary hearing loss ranged from 20% for patients where *GJB2* variants were already excluded [32] to around 40% for non-pretested patients [33].

Of the variants detected by WES as pathogenic, it was surprising to find the variants c.-22-2 and c.-22-6 in the 5′UTR of *GJB2*. It is possible that these variants affect the splicing. The variants were not detected previously by Sanger sequencing of the coding exon 2. The first 22 nucleotides of the *GJB2* exon 2 comprise the untranslated region (UTR). Consequently, this section is not usually covered by Sanger and not included in NGS for variant calling. Data on both variants as pathogenic or probably pathogenic have previously been published. Therefore, we recommend a target examination of this area with Sanger sequencing. It should be used specifically for *GJB2* patients with a heterozygous pathogenic *GJB2* variant and before the NGS of other hearing-loss genes is performed [34,35].

The spectrum of causal genes detected in the present study is broad, and with the exception of well-known genes associated with hearing loss, we detected several specific types. For the variant interpretation, we used the modified ACMG criteria for hearing loss, but most of the variants have previously been reported and proved to be pathogenic [19]. In *GPSM2*, the detected variant c.858_859delinsGT was not reported, although other variants that predicted a premature stop codon were reported as pathogenic in this gene (HGMD, accessed 02/2021). The *GPSM2* variants were reported to cause profound hearing loss DFNB82 or the Chudley–McCullough syndrome, with brain anomalies on MRI, including hypoplasia of corpus callosum. Hydrocephalus was detected in our patient, which is a condition that has previously been reported in some patients with *GPSM2* pathogenic variants [36]. The variant c.72del, p.(Ile24Metfs*22) in the *MPZL2* gene was reported in several families with the documented disease segregation in the affected members, and with the onset occurring between 3 and 9 years of age. Our patient reported the onset of hearing loss at 10 years of age. Interestingly, this variant is common among the Ashkenazi Jewish (GnomAD 0.38%) population but has also been found in Turkish families [37,38]. The pathogenic biallelic variants in the *SLITRK6* gene cause deafness and myopia. The variant c.1240C>T, p.Gln414* detected in our patient has previously been suggested to be pathogenic in Amish patients, with the variant being detected as a frequent founder variant among them. Nevertheless, the frequency of this variant in the GnomAD is reported only in the European population (non-Finnish), with a very low frequency of 0.001% [39,40]. The *POU3F4* gene is associated with DFNX2 deafness with the X-linked recessive inheritance [41], while the variant c.607_610del, p.(Gln203Glufs*37) detected in our patient is suggested to be pathogenic. The genes *GPSM2*, *MPZL2,* and *SLITRK6*, in which the homozygous rare variants were detected, indicate the high probability of the parents of the affected patients being related/distantly related people.

Variants in the *KARS1* gene cause neurological disorders other than hearing loss. We detected two missense *KARS1* variants in a compound heterozygous state in one patient with hearing loss. Only one of these variants was reported in a patient with a complicated neurological phenotype and hearing loss [42]. Since the variants are classified as VUS, we believe these *KARS1* variants could explain the hearing loss in the patient. Both were detected in the trans position and are predicted to affect the splicing; therefore, an additional functional study should be performed in order to clarify their potential pathogenicity. The absence of the second pathogenic variant in the DFNB1 region and the diagnostic yield (22%) of the WES in the patients suggests that the cause of hearing loss may be found in genes other than those related to the *GJB2* gene. A possible reason for an excess of the *GJB2* heterozygotes among hearing loss patients is the small/limited number of patients included in the previous studies, at least among Czech patients. Moreover, an excess of patients with a heterozygous pathogenic *GJB2* variant could be attributed to the limited examination of the pathogenic *GJB2* variants. These included not only the first known SNV but also the subsequently discovered deletions, del(GJB6-D13S1830), del(GJB6-D13S1854), and c.-23+1G>A in the non-coding exon 1—an examination of which was not included in the first reports. This supports the 2.2% frequency of c.35delG heterozygotes detected among the 3278 normal-hearing gamete donors. These gamete donors were tested at the Centre for Medical Genetics and Reproductive Medicine Gennet in the Czech Republic (currently unpublished data). The frequency of *GJB2* heterozygotes in healthy individuals (2.2%) is more relevant to the Czech population than the GnomAD NFE (1.9%) and is not too far from a previously found frequency of 4% among those with hearing loss. Future studies should focus on other populations and compare the number of individuals with the c.35delG monoallelic variant in those with hearing loss to normal hearing individuals.

Interestingly, results that support our findings come from a Chinese study where it was found that the c.235delC pathogenic variant is very frequent among the Chinese population [43]. In a large cohort of patients with hearing loss, 3.89% of the patients were patients with a heterozygous pathogenic *GJB2* variant yet, in the general population, the frequency was lower at 2.45%. And in 29.5% of Chinese patients, the pathogenic variants in genes other than *GJB2* were detected with WES. If we subtract these clarified patients (29.5%) from the detected frequency (3.89%), we find that the frequency of individuals with a heterozygous pathogenic *GJB2* variant with hearing loss versus the general population is almost the same (3.89–1.15%). This indicates that even in the Chinese population, an excess of patients with a monoallelic *GJB2* variant is elucidated by the cause of hearing loss in genes other than *GJB2*, thus making the occurrence of a frequent unknown variant in the DFNB1 region unlikely.

A possible limitation of our study could be the size of the selected DFNB1 region. We included the *GJB2* gene and the common interval into the design of our study, but we could have missed some unknown specific region. Nevertheless, the region examined in this study is consistent with the results of Moisan et al., who characterized the position of *GJB2* cis regulatory elements and reported that all four *GJB2* enhancers were localized in the 95.4kb common interval [44].

We could not confirm the existence of the presumed, second causal variant in the DFNB1 region that would explain the cause of hearing loss in at least a number of patients with a heterozygous pathogenic *GJB2* variant. Nevertheless, our study showed that in a substantial number of these patients, the cause of hearing loss can be clarified by pathogenic variants in genes other than the *GJB2*. We recommend commencing the Sanger sequencing of the *GJB2* gene before WES since it is possible to cover the 5´UTR of exon 2 and non-coding exon 1 with Sanger; these regions are covered poorly by WES. The MLPA method should supplement the *GJB2* examination of known CNVs. Future studies should focus on additional populations in the diagnostic yield of the WES in patients with a *GJB2* monoallelic variant. In the families of patients with a heterozygous pathogenic *GJB2* variant, the extended Sanger sequencing and WES could clarify the cause of hearing loss in 22% of the patients while enabling precise genetic counselling and an estimation of the risk of hearing loss recurrence in their relatives.

## Figures and Tables

**Figure 1 genes-12-00684-f001:**
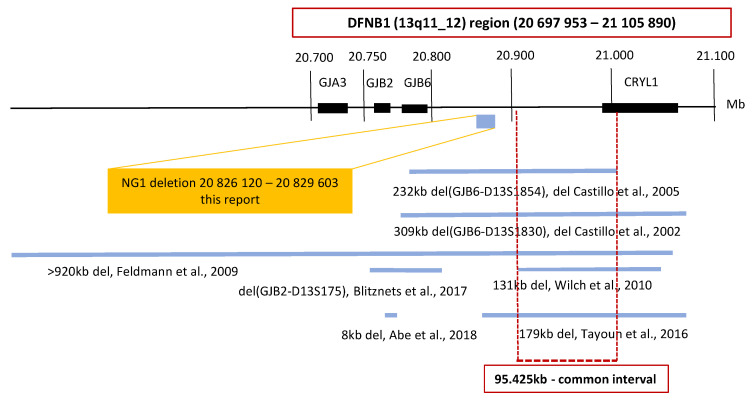
Schematic map of the DFNB1 (13q11-q12) region. The large deletions reported at the DFNB1 region are shown. The coordinates are based on the Human Genome Build GRCh37/hg19. An intersection of five of the large deletions displays a common interval. The *GJB2*/DFNB1 region examined in this study is highlighted by the red box. The included genes and the 95.425kb large common interval are shown. The only detected nonpathogenic deletion NG1 is presented in the yellow box.

**Figure 2 genes-12-00684-f002:**
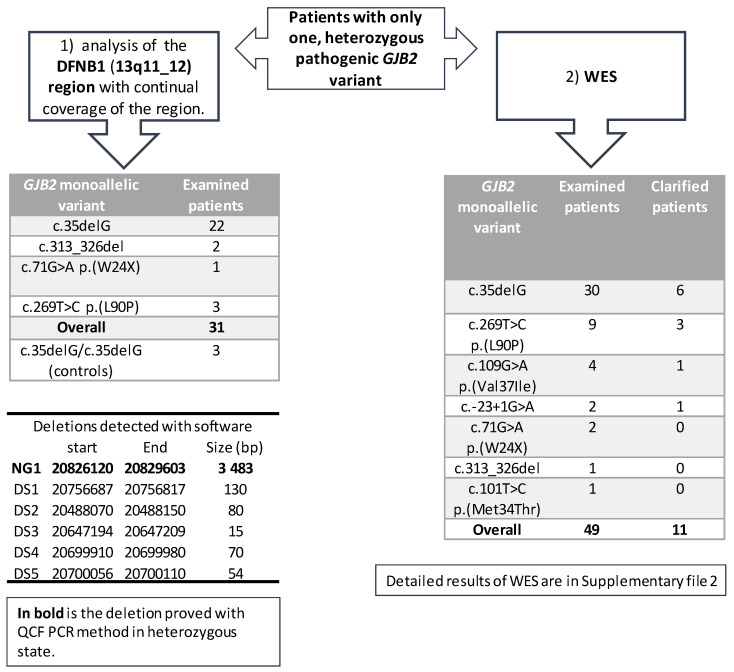
An overview of the examination of patients that are heterozygous for a *GJB2* variant. The spectrum of heterozygous (monoallelic) variants for the DFNB1 (13q11-q12) region, and WES is included. An overview of the deletions detected with different software is included.

## Data Availability

Data available upon request from the authors.

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
