# Peer review of "The Cause of Hereditary Hearing Loss in GJB2 Heterozygotes—A Comprehensive Study of the GJB2/DFNB1 Region"

_genes, 2021, doi:10.3390/genes12050684_

Round 1
Reviewer 1 Report
This is an interesting study on a long-standing problem concerning the high frequency of the so-called ‘monoallelic’ GJB2 cases, i.e. subjects with recessive hearing loss who carry only one GJB2 mutated allele. The experimental design is correct, results are sound, and the interpretation of data is appropriate. I have only some minor comments:
1) Lines 126-127. ‘Only CNV variants detected in at least three monoallelic patients, and not present in biallelic controls, were evaluated’. This approach is sensible. However, it could miss very rare deletions that may be present in just one or two patients. Given the negative results, have the authors re-examined their data to search for those hypothetical CNVs while relaxing that filter criterium?
2) Lines 281-284. Which was the degree of myopia (if any) in the subject with SLITRK6 mutations?
3) Some typos and minor edits:
a) Line 49. Insert ‘mutation’ after c.35delG.
b) Lines 51-52. Rephrase the sentence. ‘Pathogenic GJB2 variants account for 82.8% of all patients with hearing loss.’
c) Line 61 and others throughout the manuscript. No need to abbreviate the name of deletions. D13S1830 is a genetic marker, the deletion is del(GJB6-D13S1830). (same for del(GJB6-D13S1854).
d) Line 182. Insert a blank between ‘29’ and ‘Czech’.
e) Line 217. ‘has been known’ should read ‘have been known’.
f) Line 229. ‘choose’ should read ‘chose’.
g) Line 272. ‘was detect’ should read ‘was detected’.
h) Line 283. ‘p.GLN414*’ should read ‘p.Gln414*’.
i) Lines 318-319. ‘We did factor’. Rephrase it, I do not understand.
Author Response
We would like to thank the anonymous reviewer for his/her overall positive evaluation of our paper and constructive criticism. The manuscript was corrected and more information has been added. The issues raised by the referee were addressed as follows:
1) Lines 126-127. ‘Only CNV variants detected in at least three monoallelic patients, and not present in biallelic controls, were evaluated’. This approach is sensible. However, it could miss very rare deletions that may be present in just one or two patients. Given the negative results, have the authors re-examined their data to search for those hypothetical CNVs while relaxing that filter criterium?
Authors’ reply: We thank reviewer for this point. The study was designed to detect the most common CNV(s), so we aimed to detect a CNV or CNVs in at least three monoallelics. After the second part of the study, where a WES examination was performed, the data were re-evaluated for any possible CNVs not present in the controls yet no other CNV was detected. Therefore, we have changed the sentence (line134-135).
2) Lines 281-284. Which was the degree of myopia (if any) in the subject with SLITRK6 mutations?
Authors’ reply: We have limited clinical data about these patients, but we have inserted the following information: ...was detected in a patient with hearing loss who was myopic from two and a half years of age and needs to wear glasses. Lines 212-213.
3) Some typos and minor edits:
- a) Line 49. Insert ‘mutation’ after c.35delG.
Authors’ reply: Corrected and the term pathogenic variant was inserted after c.35delG in the line 50-51.
- b) Lines 51-52. Rephrase the sentence. ‘Pathogenic GJB2 variants account for 82.8% of all patients with hearing loss.’
Authors’ reply: We have corrected the sentence (line52-54)
- c) Line 61 and others throughout the manuscript. No need to abbreviate the name of deletions. D13S1830 is a genetic marker, the deletion is del(GJB6-D13S1830). (same for del(GJB6-D13S1854).
Authors’ reply: The name of both deletions was corrected to the correct term - del(GJB6-D13S1830) and del(GJB6-D13S1854) throughout the manuscript. (Lines:63, 66, 108, 329-330)
- d) Line 182. Insert a blank between ‘29’ and ‘Czech’.
Authors’ reply: Corrected.
- e) Line 217. ‘has been known’ should read ‘have been known’.
Authors’ reply: Corrected.
- f) Line 229. ‘choose’ should read ‘chose’.
Authors’ reply: Corrected.
- g) Line 272. ‘was detect’ should read ‘was detected’.
Authors’ reply: It was corrected.
- h) Line 283. ‘p.GLN414*’ should read ‘p.Gln414*’.
Authors’ reply: It was corrected.
- i) Lines 318-319. ‘We did factor’. Rephrase it, I do not understand.
Authors’ reply: We have rephrased the sentence (lines 350-351) to: We included the GJB2 gene and the common interval into the design of our study, but we could have missed some unknown specific region.
Reviewer 2 Report
Biallelic pathogenic variants in GJB2 or a pathogenic variant in GJB2 in trans with a deletion in the GJB2/GJB6 region are a frequent cause of profound hearing loss in Caucasian, Ashkenazi Jewish and East Asian populations. Here the authors aim to identify a second pathogenic allele in individuals with hearing loss with only one identified pathogenic allele in GJB2. They studied two groups of individuals with hearing loss and monoallelic pathogenic variant in GJB2, using two approaches. The genomic DNAs of 28 individuals were studied using NGS with probes that continuously cover the 0.4 Mb in the DFNB1 region for potential CNV. The genomic DNAs of 51 individuals were studied by exome sequencing to identify potential pathogenic variants.
In most cases, the authors did not identify a second pathogenic variant in the DFNB1 region. But in 20% of the individuals whose genomic DNA underwent exome sequencing, they identified another probable cause of hearing loss in other genes than GJB2. This is an important result at the time when the first clinical trials for gene therapy in the inner ear are about to start. For those to work, it is crucial to get an accurate genetic diagnosis of the hearing loss of patients, which is difficult due to the extreme genetic heterogeneity of hearing loss.
The rationale for this study is strong and will interest geneticists, clinicians and genetic counselors. The writing is generally clear although a few sentences require clarifications. Although the authors’ conclusions are well supported by the data and should still stand, these conclusions are currently biased by the inclusion of individuals with syndromic forms of hearing loss among individuals with nonsyndromic forms of hearing loss. Results should be presented only for individuals with a similar phenotype as individuals with a hearing loss caused by biallelic GJB2 pathogenic variants. Some additional analysis of the data could be done and presented here to strengthen the conclusions of this work.
The results presented by the authors are impactful for the care and genetic counseling of individuals with monoallelic pathogenic variant in GJB2.
Major comments:
Please describe the families of the patients. Do either of the parents of the individuals presented here have hearing loss too? Are there families with more than one individual with hearing loss? Were the heterozygote variants detected in other genes than GJB2 and thought to be causing the hearing loss, de novo?
Please give further details of the phenotyping of the patients and their level of hearing loss. Some of the patients don’t seem to have congenital or even prelingual forms of hearing loss (line 279). Some pathogenic variants were found in patients presenting clear syndromic forms of hearing loss rather than non syndromic forms of hearing loss. This includes a significant bias in your conclusions. Patients with clear signs leading to a diagnosis of syndromic forms of hearing loss (such as Waardenburg syndrome, growth retardation, hearing loss and pterygium colli or patients with profound hearing loss, hydrocephalus and dilated ventricular system) should not be amalgamated with patients with nonsyndromic forms of hearing loss and should not be presented together. Only patients which fulfill the criteria of GJB2 testing should be presented together (bilateral, nonsyndromic, prelingual or early hearing loss, sensorineural).
Please indicate the frequency of the variants you report in the appropriate control population(s).
Variants identified as likely causal should be deposited into accessible database for example CLINVAR.
Although reported as rare in the Czech and Slovak population, have you tested for the presence of the different known deletions in the GJB2-GJB6 region in the cohort of individuals with hearing loss studied here? This should be included and could be tested by simple PCRs.
Reference to 5 of these deletions is done in the text but 7 deletions are presented in the first figure 1. Please clarify.
Line 58 please further explain why you hypothesized that the undetected variant would be a small copy number variant (CNV)? Instead for example of a common allele which would lead to a hypomorph allele of GJB2. Such a frequent allele has been identified in patients with hearing loss and only one pathogenic variant in SLC26A4 (see PMID: 28780564). A similar analysis as reported in this publication, with the already acquired data, should be strongly considered and presented.
A discussion of other studies addressing the same critical question is missing. For example, how do your results differ from those of the article PMID: 31992338. Similar studies in population of different ancestry are valuable as results could be different in populations of different ancestry.
Other comments:
Nomenclature of the variants should be clearly indicated at both the DNA and the protein level; reference accession numbers should be indicated.
Please add OMIM ID.
In the abstract, please consider changing “the most frequent causes” for “a frequent cause” or indicating the populations in which it is the case. This is not the case in all populations.
Please consider using “hearing loss” rather than “hearing impairment” for sensitivity to the Deaf community, avoiding “disease” too and preferring “sensory defect” for example
Please consider using exome sequencing rather than whole exome sequencing as all the coding sequences of all the genes are still being identified.
Color blindness being the most common sensory defect, please change the text accordingly.
Line 43 please add which population you are referring to
Line 365 “choan atresia” should be “choanal atresia”
Please italicize all gene names.
Author Response
We would like to thank the anonymous reviewer for his/her overall positive evaluation of our paper and constructive criticism. The manuscript was corrected and more information has been added. The issues raised by the referee were addressed as follows:
Major comments:
Please describe the families of the patients. Do either of the parents of the individuals presented here have hearing loss too? Are there families with more than one individual with hearing loss? Were the heterozygote variants detected in other genes than GJB2 and thought to be causing the hearing loss, de novo?
Authors’ reply: We have expanded the description of the patients. The patients’ parents had normal hearing and the patients fulfilled the criteria for autosomal recessive inheritance; however, a sporadic occurrence could also be possible since no similarly affected sibling was present in our cohort. The segregation criteria of the detected variants, where it was possible, are presented in the Table S1.
Please give further details of the phenotyping of the patients and their level of hearing loss. Some of the patients don’t seem to have congenital or even prelingual forms of hearing loss (line 279). Some pathogenic variants were found in patients presenting clear syndromic forms of hearing loss rather than non syndromic forms of hearing loss. This includes a significant bias in your conclusions. Patients with clear signs leading to a diagnosis of syndromic forms of hearing loss (such as Waardenburg syndrome, growth retardation, hearing loss and pterygium colli or patients with profound hearing loss, hydrocephalus and dilated ventricular system) should not be amalgamated with patients with nonsyndromic forms of hearing loss and should not be presented together. Only patients which fulfill the criteria of GJB2 testing should be presented together (bilateral, nonsyndromic, prelingual or early hearing loss, sensorineural).
Authors’ reply: The overall phenotype of a majority of patients was presented with early/prelingual hearing loss and fulfilled the criteria for GJB2 testing (bilateral, nonsyndromic, prelingual or early hearing loss, sensorineural). But after an additional revision of the patients included in our study, we decided to exclude two clearly syndromic patients. A patient with a pathogenic variant in CHD7 was reported from the geneticist as a noonan syndrome and his phenotype is specific. The same can be said of the patient with the MITF variant and Waardenburg syndrome (heterochromia iridis). Nevertheless, we would like to report on a third patient with a pathogenic variant in the GPSM2 gene. He was referred to us from the geneticist as a patient with profound hearing loss that was detected at the age of two. But the patient was not reported as patient with syndromic hearing loss since, initially, hydrocephalus was not linked to hearing loss. This information was detected after we correlated the WES results with the updated clinical information. We hope the data will not be biased after these changes and still support our hypothesis.
After revising our results, we have included the patient with heterozygous VUS variants in the KARS gene among the clarified patients. Although both variants are VUS, there is a high probability, supported by in silico predictions, that they affect splicing. Moreover, genetic analysis has detected each variant coming from one parent, therefore the KARS variants are biallelic. (lines 199-201,204-206, 213-220, 265-267, 282-288, 295-298)
Please indicate the frequency of the variants you report in the appropriate control population(s).
Authors’ reply: The frequencies of the reported causal variants are very low, therefore we used the ACMG criteria classification where one of the moderate criteria is used for the frequency (PM2). Where applicable, it is presented in the table S1. The frequency is mentioned only in the specific case of the SLITRK6 variant, where the frequency is surprisingly high for a specific ethnicity and we think it is important to state it.
Variants identified as likely causal should be deposited into accessible database for example CLINVAR.
Authors’ reply: We have submitted the reported likely causal variants to CLINVAR, but the process is not directly online so it will take some time for the variants to be presented in CLINVAR.
Although reported as rare in the Czech and Slovak population, have you tested for the presence of the different known deletions in the GJB2-GJB6 region in the cohort of individuals with hearing loss studied here? This should be included and could be tested by simple PCRs.
Reference to 5 of these deletions is done in the text but 7 deletions are presented in the first figure 1. Please clarify.
Authors’ reply: In order to exclude specific deletions prior to the study, the presence of different GJB2-GJB6 deletions was examined in all the patients with a PCR test or MLPA.
We have added a sentence about this testing. (Line107-109)
The number of reported deletions was clarified in the text. (line 68-69, line 81-85)
Line 58 please further explain why you hypothesized that the undetected variant would be a small copy number variant (CNV)? Instead for example of a common allele which would lead to a hypomorph allele of GJB2. Such a frequent allele has been identified in patients with hearing loss and only one pathogenic variant in SLC26A4 (see PMID: 28780564). A similar analysis as reported in this publication, with the already acquired data, should be strongly considered and presented.
Authors’ reply: We hypothesised that CNV is one of the most likely candidates and this is supported by the common interval, where there is an intersection of the five large deletions. An additional supporting idea, is the presence of a regulatory region in the common interval where the cis regulation of the GJB2 expression has been proved, and is referred to in our introduction.
Moreover, we did a haplotype analysis of 28 patients analysed for the DFNB1 region, and the analysis showed there was a common frequent haplotype among monoallelic. No specific haplotype, for at least part of the monoallelic, was detected. Moreover, in some of the monoallelic patients with a detected common frequent haplotype, we showed, by WES, that there were pathogenic variants in genes other than GJB2. All these observations support the idea that there is no specific haplotype. In order to ensure that the content of our message was as straightforward and clear as possible, we decided not to include the haplotype analysis.
.
A discussion of other studies addressing the same critical question is missing. For example, how do your results differ from those of the article PMID: 31992338. Similar studies in population of different ancestry are valuable as results could be different in populations of different ancestry.
Authors’ reply: We thank the reviewer for the important comment. We expand on this interesting article in the discussion.(lines 339-349)
Other comments:
Nomenclature of the variants should be clearly indicated at both the DNA and the protein level; reference accession numbers should be indicated.
Please add OMIM ID.
Authors’ reply: Variants at the DNA and protein level with accompanying reference number s are included in the Table S1. But we also added the reference number and the OMIM ID for each gene to the text.
In the abstract, please consider changing “the most frequent causes” for “a frequent cause” or indicating the populations in which it is the case. This is not the case in all populations.
Authors’ reply: We have changed it.(line 19)
Please consider using “hearing loss” rather than “hearing impairment” for sensitivity to the Deaf community, avoiding “disease” too and preferring “sensory defect” for example
Authors’ reply: We have made the suggested changes throughout the manuscript.
Please consider using exome sequencing rather than whole exome sequencing as all the coding sequences of all the genes are still being identified.
Authors’ reply: We would like to keep the term WES – whole exome sequencing, as it is a well-established term even if it does not mean that all the coding sequences are already known. The specific version of WES that allowed us to identify the examined range is included in the Materials and methods section.
Color blindness being the most common sensory defect, please change the text accordingly.
Authors’ reply: We have corrected the sentence. (Line 36-37)
Line 43 please add which population you are referring to
Authors’ reply: The literature refers to the Czech population, but to be clearer from the first look, we have stated it explicitly. (line 43)
Line 365 “choan atresia” should be “choanal atresia”
Authors’ reply: This section has been deleted.
Please italicize all gene names.
Authors’ reply: All genes names should now be in italics.
Round 2
Reviewer 2 Report
We would like to thank the authors for their responses and modifications of the manuscript.
Several points still need to be addressed :
The title does not reflect the content of the article. The exact cause of hearing loss in most of these patients is still to be determined.
Please verify the percentages you indicate as they seem inconsistent between abstract and rest of the text.
Line 271 does not make sense as is, and should be corrected to 2 sentences?: “It is possible that these variants affect the splicing since they were not detected by Sanger sequencing of the coding exon 2.”
Lines 347 to 349 need clarification. Do you mean that your results are consistent with ….
Article should be reviewed for English grammar and wording before publication, in particular the discussion needs some work and could be shortened.
Supplementary files:
For patient CZ-218517 even if the parents of this individual are not available, please check that the 2 variants are compound heterozygotes. This can be easily done considering the distance between the 2 variants and should be indicated in the Supplementary file_Table1_detected variants.
Please indicate the frequency of the variants you reported in the appropriate control population(s), and the maximum frequency in any population as well as the number of homozygotes (recessive) or heterozygotes (dominant) in gnomAD for example, this is in particular important for the missense variants you identified and are thinking to be causal.
Variants identified as likely causal should be deposited into accessible database for example CLINVAR.
Please indicate the reference of your submission.
Response to review:
In response to: Line 58 please further explain why you hypothesized that the undetected variant would be a small copy number variant (CNV)? Instead for example of a common allele which would lead to a hypomorph allele of GJB2. Such a frequent allele has been identified in patients with hearing loss and only one pathogenic variant in SLC26A4 (see PMID: 28780564). A similar analysis as reported in this publication, with the already acquired data, should be strongly considered and presented.
You indicated
We hypothesised that CNV is one of the most likely candidates and this is supported by the common interval, where there is an intersection of the five large deletions. An additional supporting idea, is the presence of a regulatory region in the common interval where the cis regulation of the GJB2 expression has been proved, and is referred to in our introduction.
> Please explain why changes of nucleotides affecting this region could not have the same effect as a CNV change. Are all the genomic regions which may affect the expression of GJB2 known and limited to one region of the genome?
Moreover, we did a haplotype analysis of 28 patients analysed for the DFNB1 region, and the analysis showed there was a common frequent haplotype among monoallelic. No specific haplotype, for at least part of the monoallelic, was detected. Moreover, in some of the monoallelic patients with a detected common frequent haplotype, we showed, by WES, that there were pathogenic variants in genes other than GJB2. All these observations support the idea that there is no specific haplotype. In order to ensure that the content of our message was as straightforward and clear as possible, we decided not to include the haplotype analysis.
>This is at the center of what you present as the goal of your publication: please present this result of a common frequent haplotype among monoallelic even if this result is only found in a subset of the patients. This is an important result.
Author Response
We thank the rewiever for his/her overall positive evaluation of our paper and constructive criticism. The manuscript was corrected and more information has been added. The issues raised by the referee were addressed as follows:
The title does not reflect the content of the article. The exact cause of hearing loss in most of these patients is still to be determined.
Authors’ reply:
We thank the reviewer for this point. In our study we tried to clarify the cause of hearing loss in patients with only one pathogenic mutation in the GJB2 gene. Although we only clarified a portion of the GJB2 heterozygotes, we drew attention to the surprising conclusion that a substantial part of these patients have hearing loss because of genes other than the GJB2 and are heterozygotes only by chance. Two of these patients had a second pathogenic variant in the GJB2 gene, but outside of the coding region – in the 5´UTR. At the same time, the second part of our study points out that we have thoroughly analysed the DFNB1 region, for CNV and point variants; and, for these points, we would like to keep the title of the paper as it is.
Please verify the percentages you indicate as they seem inconsistent between abstract and rest of the text.
Authors’ reply:
We indicate the number of clarified patients due to pathogenic (causal) variants in the results - Lines 214-216. However, WES revealed the cause of hearing loss in 22% of GJB2 monoallelic patients (9 in other genes +2 in the GJB2 /47).
Line 271 does not make sense as is, and should be corrected to 2 sentences?: “It is possible that these variants affect the splicing since they were not detected by Sanger sequencing of the coding exon 2.”
Authors’ reply: The sentence is now divided in to two separate sentences.
“It is possible that these variants affect the splicing. These variants were not previously detected by Sanger sequencing of the coding exon 2.” (line 275-276)
Lines 347 to 349 need clarification. Do you mean that your results are consistent with ….
Authors’ reply:
We agree; the sentence was changed. Line 355-357.
Article should be reviewed for English grammar and wording before publication, in particular the discussion needs some work and could be shortened.
Authors’ reply:
The article has been reviewed for its spelling and use of English grammar by a UK scientist with a Ph.D in Neuroscience. All the changes, as well as the whole manuscript, have been corrected accordingly.
We have read the discussion fully. All sections of the discussion are interconnected therefore, shortening it would be, in our opinion, detrimental.
Supplementary files:
For patient CZ-218517 even if the parents of this individual are not available, please check that the 2 variants are compound heterozygotes. This can be easily done considering the distance between the 2 variants and should be indicated in the Supplementary file_Table1_detected variants.
Authors’ reply:
We thank reviewer for this remark. In reviewing the bam files of the CZ-218571 patient, it can be clearly seen that the variants are present in different reads and not in the same read. This means that these variants are in the trans position, so the patient is a compound heterozygote. We have indicated this in the family history column in Supplementary file_Table1_detected variants - compound heterozygote, variants detected in the trans position through analysis of the bam files.
Please indicate the frequency of the variants you reported in the appropriate control population(s), and the maximum frequency in any population as well as the number of homozygotes (recessive) or heterozygotes (dominant) in gnomAD for example, this is in particular important for the missense variants you identified and are thinking to be causal.
Authors’ reply:
The GnomAD allele frequencies of the NFE population, as well as the popMax frequency, are now in the Supplementary file_Table1_detected variants. Also the number of heterozygotes, as well as all the examined individuals, is indicated.
Variants identified as likely causal should be deposited into accessible database for example CLINVAR.
Please indicate the reference of your submission.
Authors’ reply: The variants were submitted with the Submission ID SUB9475289, Organization ID: 508035.Currently, we are waiting for the validation process. If the data were submitted without errors, then it will be displayed automatically in ClinVar. Once the variants are in ClinVar, we will upload the PubMed number of the article, and the data will then be available.
Response to review:
In response to: Line 58 please further explain why you hypothesized that the undetected variant would be a small copy number variant (CNV)? Instead for example of a common allele which would lead to a hypomorph allele of GJB2. Such a frequent allele has been identified in patients with hearing loss and only one pathogenic variant in SLC26A4 (see PMID: 28780564). A similar analysis as reported in this publication, with the already acquired data, should be strongly considered and presented.
You indicated
We hypothesised that CNV is one of the most likely candidates and this is supported by the common interval, where there is an intersection of the five large deletions. An additional supporting idea, is the presence of a regulatory region in the common interval where the cis regulation of the GJB2 expression has been proved, and is referred to in our introduction.
> Please explain why changes of nucleotides affecting this region could not have the same effect as a CNV change. Are all the genomic regions which may affect the expression of GJB2 known and limited to one region of the genome?
Authors’ reply:
We agree with the reviewer, that small nucleotide changes (SNV) could produce the same effect as CNV. Our study was primarily focused on a CNV analysis, followed by a SNV analysis; however, no pathogenic variant was detected (except the c.-22-6 GJB2). Lines 193-198
Moreover, we did a haplotype analysis of 28 patients analysed for the DFNB1 region, and the analysis showed there was a common frequent haplotype among monoallelic. No specific haplotype, for at least part of the monoallelic, was detected. Moreover, in some of the monoallelic patients with a detected common frequent haplotype, we showed, by WES, that there were pathogenic variants in genes other than GJB2. All these observations support the idea that there is no specific haplotype. In order to ensure that the content of our message was as straightforward and clear as possible, we decided not to include the haplotype analysis.
>This is at the center of what you present as the goal of your publication: please present this result of a common frequent haplotype among monoallelic even if this result is only found in a subset of the patients. This is an important result.
Authors’ reply:
We should clarify this discrepancy in our previous reply.
We did the haplotype analysis from the NGS data in 28 monoallelic and 3 biallelic controls. The data were assembled from 29 SNPs detected with NGS around the GJB2 gene. There was a shared haplotype in biallelic controls and also in the monoallelic patients, so it is a common – frequent haplotype in general. And it was not possible to use these analysis to pick at least some of the monoallelics to focus on a specific genetic condition (shared CNV or SNV). We have included the detected common - frequent haplotype among the monoallelic in the methods and results sections. Lines 123-125, 199-205